# The Functional and Mechanistic Roles of Immunoproteasome Subunits in Cancer

**DOI:** 10.3390/cells10123587

**Published:** 2021-12-20

**Authors:** Satyendra Chandra Tripathi, Disha Vedpathak, Edwin Justin Ostrin

**Affiliations:** 1Department of Biochemistry, All India Institute of Medical Sciences Nagpur, Nagpur 441108, MH, India; disha-vedpathak@outlook.com; 2Department of General Internal Medicine, University of Texas MD Anderson Cancer Center, Houston, TX 77030, USA

**Keywords:** ubiquitin–proteasome system (UPS), immunoproteasome, solid tumors, proteasome inhibitors

## Abstract

Cell-mediated immunity is driven by antigenic peptide presentation on major histocompatibility complex (MHC) molecules. Specialized proteasome complexes called immunoproteasomes process viral, bacterial, and tumor antigens for presentation on MHC class I molecules, which can induce CD8 T cells to mount effective immune responses. Immunoproteasomes are distinguished by three subunits that alter the catalytic activity of the proteasome and are inducible by inflammatory stimuli such as interferon-γ (IFN-γ). This inducible activity places them in central roles in cancer, autoimmunity, and inflammation. While accelerated proteasomal degradation is an important tumorigenic mechanism deployed by several cancers, there is some ambiguity regarding the role of immunoproteasome induction in neoplastic transformation. Understanding the mechanistic and functional relevance of the immunoproteasome provides essential insights into developing targeted therapies, including overcoming resistance to standard proteasome inhibition and immunomodulation of the tumor microenvironment. In this review, we discuss the roles of the immunoproteasome in different cancers.

## 1. Introduction

The ubiquitin-proteasome system (UPS) is a multicomponent, multiprotein structure that catalyzes the proteolysis of unwanted, misfolded, and foreign proteins that have been covalently modified with ubiquitin molecules [1]. Selective proteolysis performed by the UPS has been associated with almost every biological process within the cell [2]. The barrel-shaped 26S proteasome complex is composed of 20S core particles associated with two regulatory proteasome activator components. The core 20S complex is the catalytic site for protein degradation comprising multimeric subunits assembled in a ring structure [3]. Immunoproteasomes were first discovered in the early 1990s, with the observation that several proteasome subunits were induced by the pro-inflammatory cytokine IFN-γ [4,5]. Proteasomes produced with these inducible subunits carried a markedly altered catalytic activity, with increased levels of trypsin- and chymotrypsin-like activity and decreased levels of caspase-like activity [6].

The immunoproteasome carries out proteasomal degradation of protein substrates for the MHC class I restricted antigen processing pathway [7,8]. These endogenous antigenic peptides are then translocated across the ER by a transporter associated with antigen processing protein (TAP) for MHC presentation on the cell surface [9]. The MHC class I-peptide complexes are responsible for the activation of CD8+ T-cells through binding the T-cell receptor (TCR), activating the T-cell for mounting immune responses against intracellular pathogens [10]. The altered catalytic function of the immunoproteasome has been suggested to generate peptides suitable for presentation in the MHC cleft, producing peptides around 13-25 residues in length and often with hydrophobic C-termini [5,11,12]. Although the exact role of the altered catalytic function in generating these peptides is still under investigation, it is documented that a diverse range of antigenic peptides is produced through immunoproteasome activity, inducing CD8+ T-cell responses to a broad range of stimuli [13,14]. The difference in epitope generation between the constitutive and the immunoproteasome has been assigned to certain cleavage preferences for both proteasomes. This difference in substrate specificity may impact the immunopeptidome by altering the quantity of certain epitopes. This appears to be only partly explained by the increased preference of the immunoproteasome for specific P1 residues and cleavage following bulky hydrophobic amino acid residues [15]. Both proteasome isoforms also have a different production kinetics affecting quantity of epitopes [16]. Apart from its function in cell-mediated immunity, the immunoproteasome has been shown to have significant roles in inflammation, autoimmunity, and cancer. There has been an ever-growing list of novel functions of the immunoproteasome in regulating inflammatory processes, cytokine secretion, as well as facilitating protein homeostasis, cell differentiation, and cell signaling [17,18,19].

Due to its myriad functions, the immunoproteasome has become a focus in the investigation of the pathology of autoimmune conditions, cancer, inflammatory diseases, and neurodegenerative disorders (Figure 1). In tumorigenesis, there have been several reports regarding dysregulation of immunoproteasome expression and function [20]. Studies have found that tumors express immunoproteasome subunits in a dynamic fashion, which could be correlated to disease outcomes and survival [21,22,23,24,25,26,27]. Immunoproteasome inhibitors have been studied in clinical settings against solid tumors as well as hematological malignancies, but therapeutic targeting of the immunoproteasome in tumors has only shown modest success [28,29]. This lack of efficacy could be attributed to the heterogeneous nature of immunoproteasome expression in different tumors. However, the full mechanistic and functional relevance of the immunoproteasome in neoplasia remains to be understood. In this review, we aim to highlight and discuss the functional studies that have aimed to reveal the role of immunoproteasome subunits in cancer.

## 2. Structural and Functional Differences: Constitutive and Immunoproteasome

### 2.1. Composition, Assembly, and Regulation

The prominent role of the UPS in immunity has emerged over the last three decades. The genes for several components of the UPS, including TAP genes and 20S subunits, were found to be located within the genomic regions containing MHC class-II genes [30,31,32]. Soon after, the proteasome was reported to have a crucial function in antigen processing for the MHC class-I presentation [4]. Studies revealed that IFN-γ induced changes in the levels and the composition of proteasomal subunits, producing a central core with altered catalytic activity. The resulting protein complex was named the immunoproteasome to highlight its role in the processing and presentation of endogenous antigens [5,11,12,33,34]. The immunoproteasome shares structural similarities in its scaffold with the constitutive proteasome, which has alternatively been called the 26S proteasome. Its supramolecular structure is a cylindrical protein complex composed of catalytic 20S core particle (CP) and two regulatory components covering the two ends of the barrel-shaped molecule. The 20S core particle consists of two pairs of heptameric ring structures. The two inner rings are built from seven β-subunits (β1-7), with the two outer rings consisting of seven α-subunits (α1-7) [3,35]. The catalytic properties of the CP in the constitutive proteasome are attributed to the β1, β2, and β5 subunits of the inner rings, with each subunit possessing distinct proteolytic activity [36].

In the generation of the immunoproteasome, β1, β2, and β5 are replaced by more efficient IFN-γ inducible subunits, which are termed β1i, β2i, and β5i. β1i is also known as large multifunctional peptidase 2 (LMP2) and is encoded by the gene Proteasome Subunit Beta type 9 (*PSMB9*). β2i is also known as LMP10 or multi-catalytic endopeptidase complex-like-1 (MECL-1) and is encoded by the *PSMB10* gene. β5i is alternatively called LMP7 and is encoded by the *PSMB8* gene [37,38]. The outer α-rings associate with the regulatory complexes that cap the two ends of the CP to allow the entry of substrates into the catalytic core, and thus serve as proteasome activators. Typically, three complexes termed PA28 (11S proteasome activator), PA200, and PA700 (19S proteasome activator), interact with the α-subunits [39]. Similar to inducible β-subunits, inflammatory stimuli like IFN-γ induce selective association with the PA28 complex and the 20S CP to form the immunoproteasome. The PA28 regulatory complex is a heptameric protein structure, composed of two homologous α (*PSME1*) and β (*PSME2*) subunits forming a heteroheptamer. A homoheptameric variant of PA28, composed of only of one γ subunit, (*PSME3*) typically occurs in the nucleus [40,41].

The basic assembly of the 20S core particle is similar for both the constitutive and immunoproteasome [42]. Synthesis begins with the formation of the outer heptameric α-rings, assisted by the proteasome-assembling chaperons (PAC proteins). The PAC1/2 chaperones stabilize the nascent outer ring complex, while PAC3/4 facilitates the formation of the ring structure by allowing the end subunits to join [43]. β-ring assembly begins with the recruitment of the β2 subunit by PAC3, with addition to the nascent α-ring structure. Next, PAC3/4 dissociates, allowing the remaining β-subunits, in a defined order of β3, β4, β5, β6, β1, and finally β7, to become incorporated into a half-formed proteasome with the outer α-ring [43,44]. Further assembly of the 20S core particle is mediated by chaperone proteasome maturation protein (POMP or proteassemblin), which fuses the two half-proteasomes. The final assembly of the 20S core requires cleavage of N-terminal pro-peptides on the catalytic β subunits to reveal a threonine residue in the active site [44]. In cells expressing both constitutive and inducible catalytic β-subunits, the assembly of the immunoproteasome is favored over the constitutive proteasome [45]. In contrast to standard proteasome assembly, the incorporation of β1i is essential for the addition of β2i subunit [42]. Next, the incorporation of β5i facilitates the maturation of the immunoproteasome by cleaving the pro-peptides from β1i and β2i [46,47]. Biogenesis of the immunoproteasome is also dependent on POMP, which is also transcriptionally induced by IFN-γ [48]. The selective preference for the synthesis of immunoproteasome over the constitutive proteasome has been attributed to the higher binding affinity of POMP to β5i over β5 [49,50].

Incorporation of the PA28 proteasome activator to the core particle to form the proteolytically active 26S immunoproteasome is an ATP-independent process [51]. The PA28 regulatory complex can associate with the CP as a single unit capping only one side, or as a pair covering both ends, or in combination with another regulator such as PA700 on either side. Cells expressing both constitutive and immunoproteasome subunit genes may also form a hybrid between the two, which have been termed intermediate proteasomes [52,53,54]. Generally, an intermediate proteasome will be composed of only one (β5i) subunit or two subunits (β5i and β1i) which are incorporated in place of the constitutive subunits [55]. The immunoproteasomes have a relatively shorter half-life due to their transient and inducible nature [48]. Similar to their constitutive counterparts, β2i and β5i subunits exhibit trypsin-like and chymotrypsin-like enzymatic activities respectively, with the similar peptide specificity. However, the β1i subunit displays more chymotrypsin-like activity as opposed to the caspase-like activity such as the β1 subunit [56,57].

The transcriptional regulation of the immunoproteasome is mediated by multiple pathways. The IFN-γ cytokine network is the most established inducer of immunoproteasome subunits, along with other antigen processing machinery such as TAP-1, PA28, and MHC class I and class II molecules [40,58,59]. Upon activation of IFN-γ signaling, the downstream mediators signal transducer and activator of transcription-1 (STAT-1) and IFN-γ regulatory factor 1 (IRF-1) upregulate the expression of catalytic βi subunits [58,60,61]. PA28αβ is upregulated via inhibition of PA700 and its preferential incorporation into the immunoproteasome through dephosphorylation of 20S core particle, both of which are also mediated by IFN-γ [62]. Type-I interferons IFN-α and IFN-β also regulate the immunoproteasome expression, which was demonstrated after hepatitis C and coxsackievirus infection [63,64,65]. Tumor necrosis factor-alpha (TNF-α) also has been shown to upregulate immunoproteasome expression upon liposaccharide mediated inflammatory stimulus [66]. Induction of the immunoproteasome has also been observed after nitric oxide (NO) exposure, constituting a cytokine-independent regulatory mechanism. This occurs through NO-mediated activation of c-AMP/PKA axis, leading to nuclear translocation of cAMP-responsive element-binding protein (CREB), which induces immunoproteasome subunit genes [67]. Apart from cellular mediators, environmental stressors can induce immunoproteasome, for instance exposure to heat shock and H_2_O_2_ [68,69]. Metabolic signals, including hyperglycemia has also been reported to regulate immunoproteasome expression [70]. Additionally, transcription factors such as NFκB, AP-1, Sp1 and Zif268 (also known as Egr1) also control transcription of individual immunoproteasome subunits [71,72,73].

### 2.2. Functions of Immunoproteasome in Immune and Non-Immune Cells

The structural and functional properties of the inducible catalytic subunits of the immunoproteasome are modified compared to their standard proteasome counterparts to specifically generate peptides for presentation in the MHC class I cleft. The β1i subunit possesses two main amino acid substitutions, Thr21Val and Arg45Leu, compared to the β1 subunit that have been shown in crystallographic studies to minimize the size of the S1 pocket in the catalytic site and to allow small hydrophobic residues to occupy the substrate-binding channel. These alterations help to produce peptides with non-polar C termini (Leu, Ile, or Val) which fit better in the MHC class I cleft. These modifications also reduce the caspase-like activity of the β1i while promoting its chymotrypsin-like function. The changes in the β5i subunit increase the hydrophilic character of its catalytic site, generating a more favorable environment for peptide bond hydrolysis [13]. While comparison of WT with β5i and β2i double knockout (KO) mice show the importance of the immunoproteasome in the generation of abundant and diverse CD8 T cell epitopes, the loss of each immunoproteasome subunit has a rather a moderate effect on MHC class I antigen presentation [74,75,76]. However, with deficiency of two or all three βi subunits, a profound decrease in the presentation of CD8 epitopes was observed [14]. Several functional studies have reported that the immunoproteasome can facilitate stronger antigenic responses to CD8 T cell epitopes [77,78,79,80,81]. On the other hand, for some antigenic peptides, CD8 T cell epitopes can be processed through the constitutive proteasome and induce stronger cytotoxic responses than peptides generated by immunoproteasome subunits [39,82,83]. Although dendritic cells efficiently present tumor antigens via immunoproteasomes, there are few reports suggesting that antigen processed through immunoproteasome are not immunogenic as compared to constitutive proteasome, which impairs priming of T cells in melanoma patients [82,84]. However, the unique structural advantages of the immunoproteasome allow for efficient presentation of many MHC class I peptides [85,86,87]. Recently, studies have discovered a thymocyte-specific isoform of β5 subunit encoded by *PSMB11* to generate the “thymoproteasome,” which has been shown to be responsible for the presentation of self-antigens during T cell development [88]. Differential antigen processing by these proteasomes has been extensively reported [89,90,91].

Antigen processing through proteasome catalyzed peptide splicing has also been reported to contribute to the immunopeptidome [92]. These splicing reactions through proteasomes were suggested to be preferably cis-splicing, which can also be in a reverse order and an outcome of transpeptidation reaction [92,93,94,95,96,97]. Constitutive proteasome and immunoproteasome have varied effects on peptide splicing and presentation [92,98]. Liepe et.al., demonstrated that one third of total HLA class I immunopeptidome and one fourth of it represented on cell surface consist of splice peptides and is comparable to non-spliced tumor associated epitopes [99,100]. The biological relevance of these spliced peptides is still controversial. Further studies are needed to validate these findings that these epitopes do exist and are not the unclassified peptides from any novel posttranslational modification or generated from a non-canonical transcript [101].

The PA28 regulatory complex may have the ability to associate with both the standard and the immunoproteasome; however, IFN-γ stimulus induces selective binding to the immunoproteasome complex [102]. This association increases the enzymatic activity of the 26S immunoproteasomal complex dramatically compared to the constitutive proteasomal assembly, which was demonstrated by several kinetic studies [103,104,105,106,107]. PA28αβ regulates the structural conformation of α rings to allow entry of substrates and release of cleaved products but surprisingly does not alter the β catalytic units directly [108,109]. This 26S immunoproteasome complex has been reported to preferentially generate longer hydrophilic peptides [110]. PA28 deficiency has been shown to reduce MHC class I surface expression independent of other subunits, leading to a decrease in the number of epitopes presented in infected cells [111].

The immunoproteasome also directly influences T cell immunity independent of CD8 antigenic processing. Immunoproteasome-deficient T cells have been shown to have dramatically reduced expansion in response to viral infections, implying a direct role for the immunoproteasome in T cell maturation [112,113]. Moreover, immune cells deficient in *PSMB8* and other immunoproteasome subunits were reported to be incapable of producing pro-inflammatory cytokines such as IL-23, IFN-γ, IL-2, IL-4 and IL-10 [17,114,115,116,117]. The role of immunoproteasome subunits in NFκB signaling is as intriguing as it is controversial, NFκB mediated induction immunoproteasome function in inflammatory disorders has been reported as is the role of LMP2 and LMP7 in NFκB activation. [118,119]. However, the exact mechanism of NFκB regulation by immune-proteasomal degradation remains under investigation [120,121]. The immunoproteasome was also found to control Th-1 and Th-17 differentiation [18]. Through these roles, the immunoproteasome has been implicated in the pathogenesis of several inflammatory conditions and autoimmune diseases such as Hashimoto’s thyroiditis, asthma, inflammatory bowel disease, autoimmune encephalomyelitis [115,122,123,124,125,126]. Recent studies have also shown that mutations in immunoproteasome subunits are associated with the development of inflammatory conditions such as JAP (joint contractures, muscular atrophy, microcytic anemia, and panniculitis-induced lipodystrophy) syndrome, JASL (Japanese autoinflammatory syndrome) with lipodystrophy, Nakajo–Nishimura syndrome as well as several other autoinflammatory syndromes associated with proteasome dysfunction that are not necessarily limited to loss-of-function mutation in immunoproteasomal genes [127,128,129,130,131,132,133].

The inducible immunoproteasome subunits are not exclusively expressed in immune cells, but also in other tissues at basal constitutive levels, such as colon, liver, lung, kidney, and small intestine epithelium mostly in the hybrid form as intermediate proteasome [75,119,134,135,136,137,138]. Several non-immunological functions for the immunoproteasome have been described. In diabetes, dysregulation of the immunoproteasome leads to reduced cardiac muscle mass and altered skeletal muscle differentiation [19,139,140]. Studies have also shown a role for the immunoproteasome in the removal of oxidized proteins, thereby maintaining protein homeostasis upon inflammatory challenge [69,141,142,143].

## 3. Functional and Mechanistic Role of Immunoproteasome Subunits in Cancer

Neoplastic transformation is mediated by massive changes in cellular homeostasis. Induction of protein synthesis, a higher mutational burden, erroneous RNA splicing, and imbalanced redox environment due to metabolic changes all contribute to the production of misfolded or damaged proteins, requiring upregulation of protein turnover pathways [144,145]. Proteasome upregulation is a well-known contributor to tumorigenesis and was first described in breast cancer and multiple myeloma [29,146,147]. High proteasomal expression is necessary to overcome cellular stress pathways, and in some cases, to selectively degrade tumor suppressor proteins. The immunoproteasome has been shown to process tumor antigens and thereby influence both immune surveillance and immune escape (Figure 1) [83]. However, its role in tumor initiation and invasion is equivocal and the underlying mechanisms are yet to be unearthed.

### 3.1. Role of Inducible Catalytic Subunits in Cancer

The generation of MHC class-I peptides is an important facet of the maturation of cytotoxic T cells (CTLs). Given the central function of CTLs in mounting anti-tumor responses, immunoproteasome subunits induced by IFN-γ have been studied for their assumed role in cancer development. Amongst the three IFN-γ inducible β subunits, β5i has to date been most implicated in blood and solid malignancies. This subunit, encoded by *PSMB8* gene, has a wide range of expression among different cancers, which have been evaluated in non-small cell lung carcinoma (NSCLC), renal cell carcinoma, glioma, colorectal cancer, triple-negative breast carcinoma (TNBC), laryngeal, and hypopharyngeal carcinoma [21,22,148,149,150,151]. In many tumors, higher expression of *PSMB8* has been linked with poor prognosis. *PSMB8* expression was found to be upregulated in all histological sub-types of renal cell carcinoma [152]. Similarly, microarray profiling of gastric adenocarcinoma samples revealed that *PSMB8* expression in tumor tissue was associated with poor prognosis [23]. High levels of *PSMB8* are associated with more aggressive gliomas, and inhibition of *PSMB8* was shown to reduce glioma cell proliferation and migration, as well to decrease glioblastoma tumor angiogenesis [148,153]. However, the observation that high expression of *PSMB8* correlated with lower overall survival does not hold for all types of neoplasms. In NSCLC patients, high expression of *PSMB8* was frequently observed in cancers with more favorable outcomes [22]. Likewise, increased *PSMB8* expression in TNBC tumor samples was associated with better disease-free outcomes, including in those with metastatic disease [21].

The ambiguous role for *PSMB8* in oncogenesis and disease progression seems to hinge on the fact that high levels of immunoproteasome expression can facilitate or impede tumor development in different contexts. For instance, the pro-tumorigenic role of *PSMB8* in colorectal cancer is related to its role in colitis-induced chronic inflammation, which can drive neoplastic transformation of intestinal epithelium in the colon. Knockout of *PSMB8* in mice was shown to prevent colitis-associated carcinogenesis [151]. *PSMB8*-deficient mice were found to be resistant to chronic inflammation and neoplasia, with reduced expression of chemokines CXCL-1, CXCL-2, and CXCL-3. Upon induction of colitis, *PSMB8*^−/−^ mice did not show macroscopic tumor development. The authors further attributed the pro-tumor effects of PSMB8 to reduced secretion of IL-17A in inflamed colons of *PSMB8* deficient mice. The study proposed that IL-17A secretion was *PSMB8*-dependent via the NFκB signaling axis. The immunoproteasome has been shown to directly regulate NFκB signaling via direct proteolytic degradation of IκB, with knockdown of *PSMB8* preventing the nuclear translocation of NFκB [117,118]. In its role in inflammation-driven carcinogenesis, *PSMB8* serves as a promising treatment target for colorectal carcinomas. Supporting this, a study showed that ONX-914, an immunoproteasome inhibitor with a higher affinity for β5i subunit, suppressed tumor development in both preventive and therapeutic settings of colitis-induced carcinogenesis [154].

However, as mentioned, deficiency of *PSMB8* is context-dependent. *PSMB8* deficiency has been shown to promote tumor growth in a mouse model of melanoma. It has been observed that *PSMB8*^−/−^ mice implanted with B16 tumors have significant tumor growth and disease development [155]. In the absence of all three inducible subunits, mice failed to mount any anti-tumor immunity against the B16 melanoma cells, which was reflected in reduced CD8^+^ T cells in the draining lymph nodes and CTLs in the tumor microenvironment (TME) and decreased IFN-γ expression [155]. This study postulated that in melanoma carcinogenesis, IFN-γ induced immunoproteasome expression by tumor cells increases infiltration of immune cells, further adding to the pool of cytokine and chemical mediators in the TME and further upregulating IFN-γ secretion, which can exert its anti-tumor functions. IFN-γ mediated overexpression of LMP7 in melanoma cells might increase the generation of neo-antigenic peptides, further accentuating an anti-tumor response. In support of this, overexpression of *PSMB8* in melanoma cell lines increased IFN-γ secretion, leading to efficient killing of tumor cells by tumor infiltrating CTLs. This seemed to be mediated through the presentation of more diverse and immunogenic HLA-1 peptides generated through overexpression of immunoproteasome subunits [156].

Thus, reduced expression of immunoproteasome subunits is a possible immune evasion mechanism deployed by tumor cells. In lung cancer, as in melanoma, higher expression of PSMB8 is associated with a more favorable prognosis, perhaps through increased immune surveillance [22,157]. In non-small cell lung carcinoma (NSCLC), tumor cells with lower expression of immunoproteasome subunits exhibited a more mesenchymal phenotype as opposed to the epithelial morphology of NSCLC cells with higher expression levels. Along with the mesenchymal phenotype, these tumor cells possessed increased migration and invasion ability with upregulated epithelial-to-mesenchymal transition (EMT) markers. Furthermore, STAT1 signaling was inhibited via the STAT3/mTOR regulatory axis in low *PSMB8* expressing NSCLC cells. STAT1, in a mutually inhibitory relationship with STAT3, was shown to be a major downstream signaling molecule, controlling IFN-γ related genes including immunoproteasome and antigen presentation machinery. Upon treatment with IFN-γ, the mesenchymal phenotype of the tumor cells was reversed and phosphorylation of STAT1 was increased. Immunoproteasome induction in the mesenchymal-like NSCLC cell lines was shown to generate an increased diversity and quantity of MHC class I peptides. When pulsed with these generated peptides, autologous CD8 T cells demonstrated robust effector responses against tumor cells in vitro [157]. Thus, IFN-γ treatment induced immunoproteasome could potentially reverse this mechanism of tumoral immune evasion.

Immunoproteasomal subunit expression has also been reported to function as an indicator for treatment response and acquisition of chemoresistance. In both NSCLC and small cell lung cancer (SCLC), acquisition of cisplatin resistance correlated with increased expression of *PSMB8* and *PSMB9*. Treatment of cisplatin-resistant tumor cells with proteasome inhibitors led to apoptosis induction, cell cycle arrest, and mitotic catastrophe. The authors propose that upregulation of immunoproteasome expression was a response to circumvent the cellular stress induced by cisplatin treatment [158]. Sensitivity to proteasome inhibitors by tumor cells was found to be associated with immunoproteasome subunit expression. In solid and hematological tumors, cells with low expression of immunoproteasome subunits showed poor response to proteasome inhibition, with significantly lower levels of apoptosis than cells with higher expression. However, pre-exposure with IFN-γ, which favored immunoproteasome subunit expression and immunoproteasome assembly, enhanced sensitivity to proteasome inhibitors [159]. As mentioned above, this raises a possibility of induction of the immunoproteasome through the IFN-γ pathway activation as a therapeutic strategy. For instance, resistance of the proteasome inhibitor bortezomib is associated with downregulation of *PSMB8*, which can be rescued through exogenous IFN-γ, leading to resensitization [160]. The resistance to inhibitor bortezomib was also found to be associated with mutation in the PSMB8 gene loci in multiple myeloma, which further potentiates the significance of screening PSMB8 mutations as well as expression for detecting chemoresistance to therapy [161].

A similar finding was noted in breast cancer. In TNBC, sensitivity to proteasome inhibitor treatment strongly correlated with high *PSMB8* expression, with cells exhibiting UPS-driven apoptosis in response to immunoproteasome ablation [162]. To maintain high proliferative and invasive capacity, tumor cells increase protein turnover. Immunoproteasome upregulation by breast cancer cells is protective against increased proteotoxicity, which forms the part of unfolded protein response. Immunoproteasome upregulation, in this context, conceivably could be clinically targeted to overcome immunoproteasome driven chemoresistance, or could become a prognostic indicator of treatment responsiveness. Immunoproteasome expression was evaluated as a predictive marker for immune checkpoint blockade therapy in melanoma, with high expression of *PSMB8* and *PSMB9* associated with better response to anti-PD-1 and anti-CTLA-4 treatment [156].

*PSMB8* expression in tumor cells not just reprograms the cellular pathways within the cell but also affects the tumor microenvironment. In highly invasive glioblastoma, a nexus of cellular communication is maintained between tumor cells, endothelial cells, and the extracellular matrix to allow increased angiogenesis. *PSMB8* expression was reported to regulate this cellular communication. Elevated expression of *PSMB8* was found in resected glioblastomas, and inhibition of *PSMB8* reduced the migration and invasion of tumor cells in vitro. Endothelial cells demonstrated similar reduced migratory and tubulogenic properties when co-cultured with conditioned media taken from *PSMB8-*inhibited glioblastoma cell cultures. This interaction seemed to be mediated through reduced expression of vascular endothelial growth factor-A (VEGF-A) by tumor cells and integrin expression by endothelial cells. This was supported by a mouse model, which demonstrated that *PSMB8* inhibition decreased tumor vessel formation [153]. However, the mechanism of VEGF-A control by the β5i subunit remains unexplained, with the authors hypothesizing immunoproteasome mediated degradation mechanism. Other studies have reported that PSMB8 regulation of migration and proliferation in less invasive grades of gliomas was dependent on PI3K and ERK pathways [149]. In addition to transcriptional and cytokine control of *PSMB8* in cancer, regulation of *PSMB8* has been reported through microRNAs, with miR-451a shown to target *PSMB8* in prostate and thyroid cancer to prevent tumor cell proliferation and invasion [24,163].

The roles of the other two catalytic subunits, encoded by *PSMB9* and *PSMB10*, are less described in cancer. β2i, or MECL1, encoded by *PSMB10*, has been reported to be downregulated in metastatic breast carcinoma, NSCLC, and acute promyelocytic leukemia however its functional relevance in tumor development is yet to be determined [157,164]. A recent study has implicated polymorphisms in *PSMB10* as a genetic risk factor for chronic myelogenous leukemia (CML) [165]. The β1i subunit, encoded by *PSMB9*, was found to be reduced in breast cancer, renal cell carcinoma, APL, and NSCLC while elevated in melanoma and ovarian cancer [150,156,157,164,166,167,168,169]. Studies have reported a similar dichotomy as seen for *PSMB8* regarding association with overall survival. Higher expression of *PSMB9* in melanoma tumors has been linked with better patient outcomes while lower expression levels in NSCLC cells exhibited better prognosis [156,157]. A recent retrospective study on immune checkpoint therapy response for NSCLC and melanoma cohorts delineated a genetic signature of antigen processing and presentation (APM) genes which included *PSMB9*. Higher APM scores, and higher *PSMB9* expression, correlated with better responses for immune checkpoint therapy (ICB) in both NSCLC and melanoma with improved overall survival [169].

The regulation of catalytic βi-subunits in cancer is brought about by several mechanisms. As described, NFκB, mTOR, and STAT1 have been shown to regulate the expression of *PSMB8* in colon and lung cancer [154,157,158]. In acute promyelocytic leukemia (APL), the fusion transcription factor PML/RARα resulting from the causative chromosomal rearrangement (15;17) has been shown to interact with transcription factor PU.1 to repress the expression of all βi subunits [164]. As described above, contextual suppression of the immunoproteasome may provide a route for immune evasion, while upregulation may impart resistance to proteotoxicity. Aneuploidy, a common feature of neoplastic transformation, often increases protein production. Increased proteasomal degradation of tumor suppressor genes is another potential exacerbator of tumorigenesis. Constitutive proteasomal subunits are also frequently dysregulated during tumor initiation, and the induction of immunoproteasome subunits could provide extra capacity to cells undergoing intense protein turnover.

Apart from transcriptional control, immunoproteasome subunits are also regulated epigenetically. Hypomethylation of 6p21.3 CpG islands in high-grade serous epithelial ovarian carcinoma upregulates *PSMB8/9* along with antigen presentation machinery proteins. This was found to be associated with increased time until recurrence time and increased CD8 T cell infiltration [167]. Low methylation profiles were observed for *PSMB8* genomic regions in mucinous type epithelial ovarian cancers, which correlated with increased susceptibility to proteasome inhibitors [170]. Epigenetic modification of immunoproteasome subunits occurs diversely and is tumor-specific to which part of tumorigenesis it affects. Besides the regulation at the transcriptional and epigenetic level, immunoproteasome subunits themselves exhibit genetic polymorphisms which serve as susceptibility markers for certain cancers such as CML, cervical, and colon cancer [165,171,172].

### 3.2. Role of Regulatory Subunits in Cancer

Even though immunoproteasome can process varied kinds of protein substrates, association with PA28 plays an important role in the generation of CTL-specific epitopes, by alerting conformation of the α rings [173,174,175,176]. While PA28 is also inducible by IFN-γ, it is also induced upon LPS or CD40 stimulation in dendritic cells [177]. The α and β subunits of the PA28 complex are differentially expressed and regulated independently. Since studies have shown that this dynamic expression influences clinical outcomes in various cancers, it has spiked the interest as to whether the differential expression and IFN-γ independent regulation of PA28 could independently promote the generation of tumor neoantigens.

In ovarian cancer, the C-terminal fragment of PA28 (PA28S or Reg-alpha, encoded by the *PSME1* gene) was found in tumor biopsies with its presence correlated with poorer overall survival in patients, and was designated as a reliable biomarker to monitor tumor relapses and treatment [178]. Similarly, in multiple myeloma, the patients with a higher abundance of PA28α in their plasma showed reduced response to the proteasome inhibitor bortezomib [26]. Just as with the IFN-γ inducible catalytic subunits, the role of regulatory subunit expression in tumors is context-dependent. It has been reported that in oral squamous cell carcinoma (OSCC) and soft tissue leiomyosarcoma, high expression of PA28α in tumor samples corresponds with poor prognosis, while in melanoma, elevated levels of *PSME1* were associated with better overall survival [25,27,179]. In OSCC cells, inhibition of PA28α in vitro led to decreased cell proliferation and a significant reduction in invasion ability and migration, implying a role in tumor growth and metastasis [179]. A similar role was shown in breast cancer cell lines, where PA28 inhibition was shown to increase CDK15, leading to suppression of migration and invasion [180,181]. Conversely, expression of PA28β was downregulated in esophageal squamous cell carcinoma, with overexpression inhibiting tumor cell proliferation in vitro [182]. However, there are limited functional studies to understand the mechanism underlying the differential behavior of PA28 and its subunits in cancer development. A recent study highlighted individual pathways of regulation for each subunit in cutaneous melanoma. Gene set enrichment and pathway-based analysis of the individual *PSME* genes showed independent and often contrasting pathways, for instance, *PSME1* expression was positively correlated with increases in cell adhesion, apoptosis, and NFκB and Wnt signaling pathways while *PSME2* was negatively correlated with the same [25]. *PSME3* seemed to share features of *PSME1* regulation, with correlation to NFκB and Wnt signaling pathways.

Besides its role as a prognostic marker, the PA28 complex also has been studied for its feasibility as a predictive marker for treatment response. *PSME1* and *PSME2* were included as part of the APM score described above that described responsiveness to ICB in NSCLC and melanoma [169]. This finding, however, is not consistent across all immunotherapies. PA28 was found to prevent effective responses in antigen-specific immunotherapy against melanoma. The protein MART-1 (also known as Melan- A or melanoma antigen recognized by T cells) has been investigated as a potential target for immunotherapy but initial trials showed a poorer than expected immune response. In vitro studies showed that the immunodominant MART-1 epitope was not efficiently recognized by CD8 T cells, due to epitope destruction by unexpected cleavage mediated by the PA28 complex [182]. In ICB, expression of the entire PA28 was observed to be a positive response marker. Alternatively, expression of *PSME1* was found to be indicative of poor response to proteasome inhibitor treatment in relapsed or refractory multiple myeloma patients [26,169]. In another approach, PA28α was reported as an accessible target for therapeutic antibodies against prostate cancer [183].

## 4. Proteasome and Immunoproteasome Inhibitors in Cancer Therapy

Given the importance of the proteasome in many aspects of carcinogenesis, targeting proteasomal subunits with small molecule inhibitors in tumor cells has emerged as an interesting avenue for cancer treatment [184]. Numerous proteasome inhibitors have been discovered in the last 30 years, which inhibit proteasomal activity through non-covalent or covalent bonding. These two groups further contain inhibitors belonging to different chemical classes such as aldehydes, boronates, epoxyketones, α-ketoaldehyde, β-lactones, vinyl-sulfones, syrbactins, and oxathiazolones [185,186,187,188].

The United States Food and Drug Administration (FDA) has approved three proteasome inhibitors, with the first being bortezomib for multiple myeloma [189]. Bortezomib is a reversible inhibitor that binds to both constitutive as well as the immunoproteasome [190,191]. Carfilzomib, a second-generation inhibitor approved in 2012, acts on both constitutive and inducible subunits with improved efficacy over bortezomib [192]. Ixazomib is an oral proteasome inhibitor that targets only constitutive proteasome subunits in a reversible manner [193]. Proteasome inhibitors have shown promising results in a clinical setting for treating hematological cancers such as multiple myeloma [194,195]. However, for solid tumors such as TNBC, prostate, and lung cancer, proteasomal inhibition has not demonstrated the same efficacy [196,197]. The lack of response in solid tumors might be due to insufficient potency or poor tumor penetration. In a previous study, although co-inhibition of β5i and β2i subunits can induce cell death in solid tumors, the required intratumoral concentration of proteasome inhibitors was not achieved [198]. Moreover, constitutive proteasome inhibition eventually results in the acquisition of chemoresistance by tumor cells. Though the mechanism of bortezomib resistance remains unclear, it had been demonstrated that immunoproteasomal inhibition in bortezomib-resistant cells can overcome tumor relapse [160]. Hence, studies which explore agents that can coordinately inhibit both the constitutive and immunoproteasome are required.

While bortezomib possesses β5i inhibitory activity, effective achievable intratumoral concentration, drug resistance, and off-target effects prevent immunoproteasomal inhibition [199]. Carfilzomib, with its potent chymotrypsin inhibitor activity, has a higher potential of achieving this co-inhibition [200]. In pre-clinical models, carfilzomib has shown broad anti-tumor activity against NSCLC and SCLC in a synergistic effect with cisplatin [201]. Specific immunoproteasome inhibitors (IPIs) are currently in development (Table 1). PR-924 is a recently developed epoxyketone small molecule inhibitor that binds specifically to β5i [28]. PR-924 inhibited growth and proliferation of multiple myeloma cells in pre-clinical models and induced apoptosis in leukemia cell lines [28,202]. Another small molecule epoxyketone inhibitor, ONX-0914 (also known as PR-957) was found to be potent at targeting β5i and effective against bortezomib-resistant myeloma and colitis-induced colorectal cancer [203,204,205]. M3258 is a relatively new reversible inhibitor highly selective for β5i subunit. Orally bioavailable, this inhibitor has demonstrated significant efficacy in multiple myeloma xenograft models as well as higher anti-tumor activity compared to other non-selective IPIs like bortezomib in in-vivo settings. Thus, promising preclinical profile of M3258 has propelled its entry into phase I clinical trials [206,207].

Unlike the previous epoxyketone-derived IPIs, UK-101 was reported to selectively inhibit the catalytic β1i subunit and showed robust activity against prostate cancer in both in vitro and in vivo studies [208,209]. Another β1i inhibitor, IPSI-001 was found to be promising against myeloma [210]. Due to their high selectivity and lower toxicity, immunoproteasome-specific inhibitors have been touted as novel anti-cancer therapeutics. Interestingly, multiple myeloma-cells resistant to constitutive proteasome inhibitors have been shown to be better responders to IPI treatment when tumor cells were pre-exposed to them, which may indicate synergy of dual inhibition of constitutive and immunoproteasomes [211]. However, as IPIs show inhibition of both constitutive and immunoproteasome enzymatic activity, further study is required to evaluate the role of immunoproteasome inhibition alone [200,203,208]. Emergence of chemoresistance against IPIs also requires further investigation. These studies, while preliminary, highlight the potential of therapeutic targeting of immunoproteasome.

**Table 1 cells-10-03587-t001:** Immunoproteasomal subunit in different cancers: Expression, Function, and Intervention.

Immunoproteasome Subunit	Cancer	Expression in Cancer Cells	Clinical Outcome	Regulatory Mediator(s)	Use as a Functional Parameter	Therapeutic Intervention	References
PSMB8 (β5i subunit)	NSCLC	downregulated	Poor prognosis	STAT3-mTOR mediated inhibition of STAT-1	NS	IFN-γ treatment to induce IP expression	[22,157]
	Renal cell carcinoma	upregulated	Poor prognosis	NS	Prognostic biomarker	NS	[152]
	TNBC	upregulated	Survival	NS	NS	NS	[21,162]
	Glioma	upregulated	Poor prognosis	NS	NS	NS	[148,153]
	Laryngeal carcinoma	upregulated	NS	Non-receptor tyrosine kinase encoded by oncogene c-Abl	NS	Tyrosine kinase inhibitor: Nilotinib	[149]
	Hypopharyngeal carcinoma	upregulated	NS	Non-receptor tyrosine kinase encoded by oncogene c-Abl	NS	Tyrosine kinase inhibitor: Nilotinib	[149]
	Colorectal carcinoma	upregulated	Poor prognosis	Transcription factor: NFκB Proinflammatory cytokines (IL17a) and chemokines (CXCL-1/2/3) SNP encoding LMP7-K allele	NS	Inhibition of β5i with ONX-912	[151,154,172]
	Gastric adenocarcinoma	upregulated	Poor prognosis	NS	Prognostic biomarker	NS	[23]
	Melanoma	upregulated	Survival	Cytokine: IFN-γ	Prognostic biomarker, predictive marker for ICB therapy	Overexpression of β5i via IFN-γ treatment	[156]
	Prostate cancer	upregulated	NS	miR-451a	NS	NS	[24]
	Papillary thyroid cancer	upregulated	NS	miR-451a	NS	NS	[163]
	Multiple myeloma	upregulated	NS	NS	NS	Selective inhibitors: PR-924, ONX-0912	[28,204]
	Cervical cancer	NS	High risk	SNP	NS	NS	[172]
PSMB9 (β1i subunit)	Renal cell carcinoma	downregulated	NS	NS	NS	NS	[168]
	Metastatic breast carcinoma	downregulated	NS	NS	NS	NS	[166]
	NSCLC	downregulated	Poor prognosis	STAT3 -mTOR mediated inhibition of STAT-1	Predictive marker for ICB therapy	NS	[156,169]
	APL	downregulated	NS	NS	NS	NS	[164]
	Melanoma	upregulated	Survival	NS	Predictive marker for ICB therapy	NS	[169]
	Ovarian carcinoma	upregulated	NS	Hypomethylated CpG islands of 6p21.3	NS	NS	[167]
	Cervical cancer	NS	High risk	SNP	NS	NS	[172]
	Prostate cancer	upregulated	NS	NS	NS	Selective inhibitor: UK-101	[208]
	Multiple myeloma	upregulated	NS	NS	NS	Selective inhibitor: IPSI-001	[210]
PSMB10 (β2i subunit)	Metastatic breast carcinoma	downregulated	NS	NS	NS	NS	[166]
	APL	downregulated	NS	Transcription factor: PU.1	NS	NS	[164]
	NSCLC	downregulated	Survival	NS	NS	NS	[157]
	CML	NS	High risk	SNP	NS	NS	[149]
PSME1 (PA28-α subunit)	Multiple myeloma	upregulated	NS	NS	Biomarker for bortezomib treatment	NS	[26]
	OSCC	upregulated	Poor prognosis	NS	Prognostic marker	NS	[179]
	Soft tissue leiomyosarcoma	upregulated	Poor prognosis	NS	Prognostic marker	NS	[27]
	Skin cutaneous melanoma	upregulated	Survival	NS	Prognostic marker	NS	[25]
	Ovarian cancer	upregulated	Poor prognosis	NS	Biomarker for tumor relapse	NS	[178]
	NSCLC	upregulated	Survival	NS	Predictive marker for ICB therapy	NS	[169]
	TNBC	upregulated	NS	CDK15	NS	NS	[180]
	Melanoma	upregulated	Survival	NS	Predictive marker for ICB therapy		[169]
PSME2 (PA28-β subunit)	ESCC	downregulated	NS	NS	NS	NS	[181]
	Ovarian cancer	upregulated	Poor prognosis	NS	Biomarker for tumor relapse	NS	[178]
	Melanoma	upregulated	Survival	NS	Predictive marker for ICB therapy	NS	[169]
	TNBC	upregulated	NS	CDK15	NS	NS	[180]
	NSCLC	upregulated	Survival	NS	Predictive marker for ICB therapy	NS	[169]

NS: not studied, NSCLC: non-small cell carcinoma, TNBC: triple negative breast carcinoma, ICB: immune checkpoint therapy APL: acute promyelocytic leukemia, CML: chronic myelogenous leukemia, SNP: single nucleotide polymorphism, OSCC: oral squamous cell carcinoma, ESCC: esophageal squamous cell carcinoma.

## 5. Summary and Conclusions

The dysregulation of immunoproteasome expression in cancer is a well-known phenomenon, and the underlying molecular mechanisms of this have been revealed to be diverse and context-dependent. The immunoproteasome itself has diverse functions with outputs in a wide range of cellular processes. These effects may explain why immunoproteasome induction and inhibition have contradictory roles in different cancers, which mirrors the context-dependent role of inflammation in cancer. However, given the appropriate tumoral and clinical context, the immunoproteasome remains an attractive target, with *PSMB8* identified as having particular centrality [154,155,157]. Nonetheless, there is still a need to conduct deeper functional and mechanistic studies for the other catalytic and regulatory subunits, especially as several subunits have been shown to act as prognostic markers in a variety of tumors [179,180]. Small molecule therapeutics targeting immunoproteasome subunits have thus received attention as a novel class of anti-cancer drugs. The central role of the immunoproteasome in a wide variety of tumor cell and microenvironmental pathways shows its promise as a target for cancer therapy. While clinical success has only been shown for a handful of IPIs, more detailed mechanistic evaluation, with a firm eye towards tumor and inflammatory context, holds tremendous potential.

## Figures and Tables

**Figure 1 cells-10-03587-f001:**
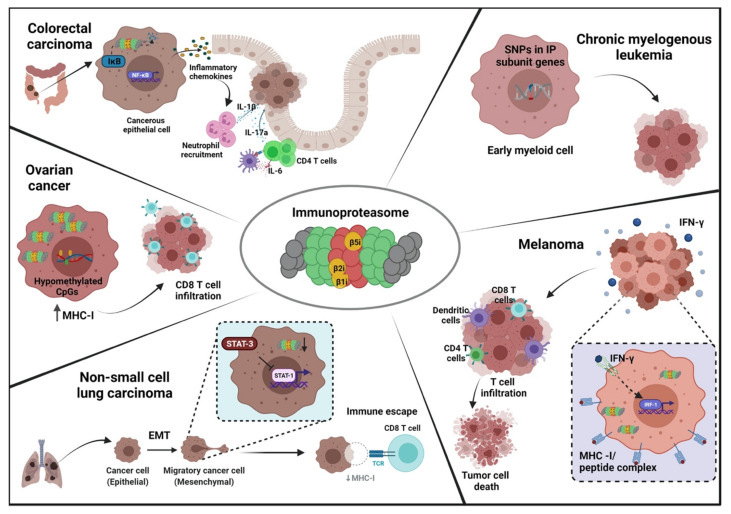
Schematic overview depicting various mechanisms of immunoproteasome participation in different cancers. Immunoproteasome can promote or inhibit tumorigenesis in various cancers through distinct and often contradictory mechanisms. In the colon, immunoproteasomal mediated degradation of IκB allows the generation of pro-inflammatory signals that eventually lead to neoplastic transformation of colonic epithelial cells. In melanoma, the inflammatory stimulus of IFN-γ increases the tumor antigen presentation and T cell infiltration, culminating in tumor cell death. In chronic myelogenous leukemia, the early myeloid cells have increased susceptibility to CML if they possess SNPs in the immunoproteasome subunit genes. In ovarian cancer cells, epigenetic modification of CpG islands promotes CD8 T cell migration into the tumor and induces CTL-mediated tumor killing. In non-small cell lung carcinoma, EMT is responsible for reducing immunoproteasome expression, thereby facilitating immune escape due to loss of MHC class I antigen presentation. EMT: epithelial to mesenchymal transition, IP: immunoproteasome, CTL: cytotoxic T lymphocytes.

## Data Availability

Not applicable.

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
