# Peer review of "The Functional and Mechanistic Roles of Immunoproteasome Subunits in Cancer"

_cells, 2021, doi:10.3390/cells10123587_

Round 1

Reviewer 1 Report

In this paper, Chandra Tripathi and colleagues propose a review of the current literature on the potential role of immunoproteasomes in tumorigenesis. The manuscript is divided into 4 main parts with the first two sections giving general insights into the structure, regulation and functions of immunoproteasomes. The last two chapters are more specifically dedicated to the relevance of the inducible subunits as (i) tumour biomarkers, (ii) tumour drivers and (iii) therapeutic targets in cancer.

Given the high number of conflicting reports on the implication of immunoproteasomes in the regulation of diverse cellular processes, it is difficult to keep a clear picture of the precise function the inducible subunits, and as such, any regular update on the latest research in this field should be welcome. The manuscript in its present form requires however major clarifications and/or modifications, as discussed below.

Major points

General comment: it is regrettable that the detrimental role of immunoproteasomes in the presentation of tumour antigens (PMID: 10661410, PMID: 16707475, PMID: 16393993, PMID: 26399368, etc.) is not better covered in this review. This point should be better addressed and discussed in this work, as it represents a major evasion mechanism for tumour cells overexpressing the inducible subunits.

General comment: the authors failed to present the ability of proteasomes to produce spliced tumour peptides (PMID: 15001714, PMID: 14724640, PMID: 16960008, PMID: 27049119, PMID: 24453253, PMID: 21670269). Given that one-third of MHC class I-restricted peptides are neoepitopes composed of two different fragments from peptide sequences which are not contiguous in the parent antigen (PMID: 27846572), the authors should address this point and discuss the role of immunoproteasomes in this process. In addition, they should emphasize that the biological relevance of these epitopes is still a matter of debate.

Page 1: “Proteasomes produced with these inducible subunits carried a markedly altered catalytic activity, with increased levels of trypsin- and chymotrypsin-like activity and decreased levels of caspase-like activity”. There is no reference for this assumption which is not entirely correct, as the chymotrypsin-like activity of Beta5i is actually lower than that of Beta5 in mice (PMID: 8612775).

Page 2: “it is clear that a diverse range of antigenic peptides is produced through immunoproteasome activity”. This assumption is based on old publications and not likely to be true. Recent investigations have shown that standard proteasomes and immunoproteasomes exhibit only minimal differences in their cleavage specificities (PMID: 25231383). The differences in the peptide repertoires generated by standard and immunoproteasomes are mostly quantitative (and not qualitative) and due to the fact immunoproteasomes exhibit a higher cleavage which, depending peptide sequence, results in increased epitope generation or destruction. This point should be clarified.

Page 2: the title of Fig.1 is unclear, please rephrase.

Page 3: “β1i is the first subunit to be recruited to the nascent immunoproteasome [42]”. This sentence is misleading, as it precludes that β1i may be incorporated together with β2 and β5, which is not the case. The authors should emphasize that β5i is required for the incorporation of β1i, which itself is a prerequisite for the incorporation of β2i.

Page 3:” Typically, two complexes termed PA28 (11S proteasome activator) and PA700 (19S proteasome activator), interact with the α-subunits”. The authors failed to comment on PA200 (PSME4) as a proteasome regulatory complex in this part of the manuscript.

Page 4: “Generally, an intermediate proteasome will be composed of only one β5i subunit or two β5i and β2i subunits which are incorporated in place of the constitutive subunits [51]”. This is not correct, β5i is mostly incorporated together with β1i and not β2i (PMID: 20937868).

Page 5: “these immunoproteasome-mediated immunological functions could be mediated entirely through modulation of NFκB signalling [102–104”]. The citing sources here do not support the assertion made by the authors. Pioneering work of Visekruna et al. (PMID: 17124531) should be mentioned here. The authors should also discuss that the impact of the inducible subunits on NF-κB signalling remains highly controversial.

Page 5: “Recent studies have also shown that mutations in immunoproteasome subunits are associated with the development of inflammatory conditions such as JAP (joint contractures, muscular atrophy, microcytic anaemia, and panniculitis-induced lipodystrophy) syndrome, JASL (Japanese autoinflammatory syndrome) with lipodystrophy and Nakajo-Nishimura syndrome[112–114]”. The authors failed to cite the following works in this section (PMID: 31783057, PMID: 26524591) and should emphasize that proteasome-associated autoinflammatory syndromes are not necessarily restricted to loss-of-function mutations in genes encoding immunoproteasome subunits (PMID: 26829627, PMID: 29805043, PMID: 30664889).

Page 5: ”The immunoproteasome is also not exclusively expressed in immune cells, but also in normal tissues at basal constitutive levels, such as colon, liver, lung, kidney and small intestine epithelium[115–120].” Incorrect. Most of these tissues expressed intermediate-type proteasomes carrying consisting of β5i/β1/β2 or β5i/β1i/β2 (PMID: 20937868). This point should be clarified.

Page 7: ”For instance, resistance of the proteasome inhibitor bortezomib is associated with downregulation of PSMB8, which can be rescued through exogenous IFN-γ, leading to resensitization[142].” Another important resistance mechanism comes from the acquisition of mutations preventing the binding of small-molecule inhibitors to the β5i active site (PMID: 31572666)

Page 8 “Immunoproteasome upregulation by breast cancer cells is protective against increases in proteotoxicity caused by the unfolded protein response”. Unclear what the authors mean here. The UPR is not viewed as toxic -when transient- and actually protects the cell from proteotoxic stress by upregulating chaperones, ERAD components, etc. This point should be clarified.

Page 8. ”The roles of the other two catalytic subunits, encoded by PSMB9 and PSMB10, are less described in cancer”. This is likely due to the fact that these are less expressed than β5i (PMID: 31572666).

Page 9: “Even though immunoproteasome can process varied kinds of protein substrates, association with PA28 plays an important role in the generation of CTL-specific epitopes, by alerting conformation of the α rings. References supporting this assumption are missing.

Page10: “In vitro studies showed that the immunodominant MART-1 epitope was not efficiently recognized by CD8 T cells, due to epitope destruction by unexpected cleavage mediated by the PA28 complex [159].” Impaired MART1 26-35 presentation is impaired not only by PA28 but by β5i and β2i as well (PMID: 26399368). This point should be emphasized and further discussed (see general comment 1).

Minor points:

Page 2 Studies have found that tumors

Page 4: due to their transient

Page 6: in TNBC tumor samples was associated with

Author Response

We would like to thank the reviewer for the constructive inputs for our manuscript. We have revised the manuscript as per the suggestions and hope that it will be as per the satisfaction of the reviewer and the editor. Please find below the answers to the comments raised by the reviewer.

Reviewer 2 Report

The review by Tripathi and co-authors thoroughly discusses the role of immunoproteasome in different cancers. I must admit that this review was a nice read from start to beginning as it represents a condensed version of all papers published in this immunoproteasome subfield. I find it very important not only for myself but also for others who follow the field to have a complete overview of differential roles of immunoproteasome in the context of different cancers.

The paper is well-structured with mostly very up-to-date references. Therefore, I suggest publication of this review as a part of the Cells Special Issue which is truly turning out to represent a very valuable selection of papers that will gather ‘all-you-need-to-know’ information on immunoproteasome.

I do have some comments, which are mostly of minor nature, but will, hopefully, contribute to further improving the manuscript.

  1. Why are some words in the title capitalized and others are not?
  2. The caption to Figure 1 seems to have a mistake. Should the word ‘participates’ be ‘participation’?
  3. In Section 2, the abbreviation for the UPS appears again. It is not needed to write ‘ubiquitin-proteasome system’ as this was abbreviated in Section 1 already.
  4. Page 3, paragraph 2, last sentence: ‘of only of one’ should be ‘of only one’.
  5. Page 4, paragraph 3: When abbreviating ‘TNF-α (tumor necrosis factor-alpha)’ and ‘CREB (cAMP-responsive element-binding protein)’, the actual abbreviations should be in parentheses.
  6. Page 4, last line: ‘chymotrypsin’ should be ‘chymotrypsin-like’.
  7. Page 5, paragraph 1, line 5: ‘a rather a moderate’ seems wrong. Please correct to ‘a rather moderate’.
  8. Given that Section 2.2. is almost exclusively focused on immunoproteasome, I suggest modifying its title to: ‘Functions of immunoproteasome in immune and non-immune cells’.
  9. Cytotoxic T cells appear to be first abbreviated on page 6, paragraph 2. However, the same term is used in page 5 already (in abbreviated form). Please take care that expressions that need abbreviations are indeed abbreviated at their first appearance in the text.
  10. Page 7, paragraph 3, last sentence: please correct ‘Thus, immunoproteasome induction, perhaps through treatment with IFN-γ treatment, …’
  11. Page 8, first paragraph: abbreviate ubiquitin proteasome system.
  12. Page 8, paragraph 2, line 14: delete the word ‘an’.
  13. Page 8, paragraph 3: define ICB.
  14. Page 9, paragraph 1, line 3: replace ‘are’ with ‘is’.
  15. Page 9, paragraph 4: ‘elevated levels of PSME1 was associated’ should be ‘elevated levels of PSME1 were associated’.
  16. Page 10, paragraph 2: ‘…on many aspects…’ should be ‘…in many aspects…’.
  17. Page 10, paragraph 3: The statement that bortezomib binds to both β5 and β5i subunits with greater affinity than other subunits is not true. Bortezomib inhibits β1i in the exact same range (single digit nanomolar IC50 value), whereas β1 is inhibited with an IC50 value of around 70 nM.

To add to this, I would modify the text regarding carfilzomib and state that carfilzomib acts on β5 and β5i subunits with improved efficacy and selectivity over bortezomib.

  1. Page 10, paragraph 3: use abbreviation for triple-negative breast cancer.
  2. Page 10: PR-957 is also known as ONX-0914 and not ONX-0912.
  3. Page 10: The authors should expand the section on immunoproteasome inhibitors. It is imperative to include compound M3258 (see J. Med. Chem. 2021, 64, 10230−10245 and Mol Cancer Ther 2021;20:1378–87). Despite not absolutely necessary, it would make sense to note that several other very selective β5i inhibitors were published recently (J. Med. Chem. 2019, 62, 7032−7041, PNAS 2016, 113 (52), E8425-E8432; Angew. Chem. Int. Ed. 2016, 55, 5745 –5748; Cui, H. et al. ChemBioChem 2017, 18, 523-526; however, most of those were evaluated in the sense to obtain compounds that would tackle autoimmune and inflammatory disease and were therefore not evaluated as potential anti-cancer drugs. Nevertheless, the authors could make a case to encourage researchers who follow this field to evaluate these β5i-selective compounds in various cancer models too.
  4. Correct references 130, 133, 136, 142, 145, 147, 148, which are missing the Journal, Issue number and pages.

Author Response

(The authors gave the same response as above.)

Round 2

Reviewer 1 Report

The authors successfully addressed all my concerns in this new version. I am pleased to recommend this work for publication.